



# Integrated water vapour content retrievals from ship-borne GNSS receivers during EUREC[4]A

Pierre Bosser[1], Olivier Bock[2, 3], Cyrille Flamant[4], Sandrine Bony[5], and Sabrina Speich[6]

[1]Lab-STICC/PRASYS UMR 6285 CNRS, ENSTA Bretagne / HOP, Brest, France
[2]Université de Paris, Institut de physique du globe de Paris, CNRS, IGN, Paris, France.
[3]ENSG-Géomatique, IGN, F-77455 Marne-la-Vallée, France.
[4]LATMOS/IPSL, UMR 8190 CNRS, SU-UVSQ, Paris, France.
[5]LMD/IPSL, UMR 8539 CNRS, Sorbonne Université, UPMC, Paris, France
[6]LMD/IPSL, UMR 8539 CNRS, ENS-Ecole Polytechnique-SU, Paris, France.

**Correspondence:** Pierre Bosser (Pierre.Bosser@ensta-bretagne.fr)

**Abstract.** In the framework of the EUREC[4]A (Elucidating the role of clouds-circulation coupling in climate) campaign that took place in January and February 2020, integrated water vapour (IWV) contents were retrieved over the open Tropical Atlantic Ocean using Global Navigation Satellite Systems (GNSS) data acquired from three research vessels (R/Vs): R/V Atalante, R/V Maria S. Merian, and R/V Meteor. This paper describes the GNSS processing method and compares the GNSS

IWV retrievals with IWV estimates from the European Center for Medium-range Weather Forecast (ECMWF) fifth ReAnalysis (ERA5), from the Moderate-Resolution Imaging Spectroradiometer (MODIS) infra-red products, and from terrestrial GNSS stations located along the tracks of the ships. The ship-borne GNSS IWVs retrievals from R/V Atalante and R/V Meteor compare well with ERA5, with small biases (-1.62 $\mathrm{kg\,m^{-2}}$ for R/V Atalante and +0.65 $\mathrm{kg\,m^{-2}}$ for R/V Meteor) and a root mean square (RMS) difference about 2.3 $\mathrm{kg\,m^{-2}}$. The results for the R/V Maria S. Merian are found to be of poorer quality,

with RMS difference of 6 $\mathrm{kg\,m^{-2}}$ which are very likely due to the location of the GNSS antenna on this R/V prone to multipath effects. The comparisons with ground-based GNSS data confirm these results. The comparisons of all three R/V IWV retrievals with MODIS infra-red product show large RMS differences of 5-7 $\mathrm{kg\,m^{-2}}$, reflecting the enhanced uncertainties of this satellite product in the tropics. These ship-borne IWV retrievals are intended to be used for the description and understanding of meteorological phenomena that occurred during the campaign, east of Barbados, Guyana and northern Brazil. Both the raw

GNSS measurements and the IWV estimates are available through the AERIS data center (https://en.aeris-data.fr/). The digital object identifiers (DOIs) for R/V Atalante IWV and raw datasets are https://doi.org/10.25326/71 (Bosser et al., 2020a) and https://doi.org/10.25326/74 (Bosser et al., 2020d), respectively. The DOIs for the R/V Maria S. Merian IWV and raw datasets are https://doi.org/10.25326/72 (Bosser et al., 2020b) and https://doi.org/10.25326/75 (Bosser et al., 2020e), respectively. The DOIs for the R/V Meteor IWV and raw datasets are https://doi.org/10.25326/73 (Bosser et al., 2020c) and https://doi.org/10.

25326/76 (Bosser et al., 2020f), respectively.



## 1 Introduction

Precise positioning with Global Navigation Satellite Systems (GNSS), in particular on the vertical component, requires the estimation of propagation delays due to the transit of the signals transmitted by the satellites through the atmosphere. These delays depend in particular on the water vapour content which is mainly located in the troposphere. As part of the GNSS data processing, the tropospheric propagation delay is modeled by a zenith component, so-called the Zenith Tropospheric Delay (ZTD), that is projected onto the receiver-satellite line-of-sight using mapping functions. Horizontal North-South and East-West gradients are also used to describe the azimuthal asymmetry of the delay. The integrated water vapour (IWV) contents are derived from the ZTD estimates. Since the late 1990s, both GNSS ZTD and IWV products have progressively been incorporated into the array of meteorological observation techniques used for atmospheric studies and are assimilated in Numerical Weather Prediction (NWP) models (Poli et al., 2007; Guerova et al., 2016). The GNSS technique possesses numerous advantages compared to other passive remote sensing techniques: the instrumentation is low-cost and power-efficient; the measurements are obtained in all weather conditions and do not require instrumental calibrations; the IWV data can be retrieved at high frequency (typically every 5 min). The agreement between GNSS-derived IWVs and their counterparts observed with more conventional meteorological instrumentation (e.g. radiosondes, microwave and infrared radiometers, lidars) is widely confirmed (Bevis et al., 1992; Haase et al., 2003; Bosser et al., 2010; Bock et al., 2013) and the accuracy of the technique is evaluated to be around 1-2 $\mathrm{kg\,m^{-2}}$ (Bock et al., 2013; Ning et al., 2016). The use of ground-based GNSS-derived IWVs for atmospheric processes studies has thus become common practice in meteorological campaigns (Haase et al., 2003; Bock et al., 2008, 2016; Hadad et al., 2018).

Since the mid-2000s, various studies have been carried out to evaluate IWV retrievals from ship-borne GNSS receivers. In this configuration, the analysis of GNSS data is more complex than for data from static terrestrial GNSS receivers due to the strong correlation between positions and propagation delays estimated with the same temporal sampling (30-300 s). Two strategies can be applied for the precise processing of GNSS data: relative positioning, which requires the use of nearby ground reference stations and from which the position of the antenna is determined relative to these reference stations, and absolute positioning for which the position of the antenna is determined directly relative to the satellites. Over open oceans, the extended distance to terrestrial reference stations prevents from using the more precise relative positioning. Absolute positioning, also called kinematic PPP (Precise Point Positioning), is mandatory there. Despite these limitations, the quality of sea-borne IWV retrievals is promising, even though it is still lower than that obtained for terrestrial stations. Compared to conventional meteorological instruments, the root mean square (RMS) of differences generally varies between 2 and 3 $\mathrm{kg\,m^{-2}}$ (Fujita et al., 2014; Shoji et al., 2017; Wang et al., 2019; Liu et al., 2019), while the RMS of differences with numerical weather prediction models ranges from 1 to 3 $\mathrm{kg\,m^{-2}}$ (Boniface et al., 2012; Wang et al., 2019; Fourrié et al., 2019).

In the framework of the EUREC[4]A (Elucidating the role of clouds-circulation coupling in climate) campaign that took place in January and February 2020, we took advantage of the presence of GNSS receivers onboard three of the research vessels (R/Vs) involved (namely, the French R/V Atalante, and the German R/V Maria S. Merian and R/V Meteor) to exploit the raw GNSS data for meteorological purposes. The three R/Vs were deployed as part of a huge experimental set up in the Tropical



West Atlantic Ocean that gathered airborne, sea-borne and island-based measuring platforms from Europe, the United States of America and the Caribbean. The objective is to provide benchmark measurements of clouds and of their environment in the trade winds and to test hypothesized cloud-feedback mechanisms thought to explain large differences in model estimates of climate sensitivity (Bony et al., 2017). During the campaign, R/V Meteor operated mainly East of Barbados documenting

atmospheric conditions upwind of the Barbados Cloud Observatory (BCO), (Stevens et al., 2016) in the so-called Trade-wind Alley. In the meantime, R/V Maria S Marian and R/V Atalante operated mainly southeast of Barbados, off the coast of Guyana and northern Brazil.

In Section 2, we present the collection of GNSS measurements gathered from the three R/Vs and the strategies used for processing the data. In Section 3 we evaluate the processing outputs from two different GNSS processing software packages. The

comparison of two software packages is motivated by the difficulty encountered in processing the lower quality data acquired from the R/V Maria S. Merian. In Section 4, we compare the GNSS-derived IWV data with those from the European Center for Medium-range Weather Forecast (ECMWF) fifth ReAnalysis (ERA5), from the Moderate-Resolution Imaging Spectroradiometer (MODIS), as well as from a set of terrestrial GNSS stations located along the tracks of the three ships. In Section 5 we draw the main conclusions regarding the processing and analysis of the ship-borne GNSS datasets.

## 2 GNSS measurements and data processing

### 2.1 GNSS measurements

The antenna, receivers and logging systems used on the three R/Vs are presented in Table 1. All three instrumentation systems were able to provide high-quality carrier phase data necessary to retrieve accurate positions and tropospheric parameters. The data logging methods differed from one R/V to another which is not a problem as all data were post-processed after the

campaign. On R/V Atalante, data were automatically saved hourly on the onboard data storage system. On R/V Maria S. Merian, a dedicated device was deployed to save data in real time and to upload the data hourly via the Internet network. Finally, on R/V Meteor, the data were saved hourly on a removable medium connected to the receiver that was retrieved at the end of the campaign. On all the receivers the measurements were made at a rate of 1 s for GNSS satellites above a minimum elevation cut-off angle of 3° for R/V Maria S. Merian and R/V Meteor and 5° for R/V Atalante. For the former two

R/Vs, only Global Positioning System (GPS) measurements were saved, while for R/V Atalante both GPS and GLONASS (Globalnaïa Navigatsionnaïa Spoutnikovaïa Sistéma) measurements were saved. However, the sake of homogeneity between the three datasets, only data from the GPS constellation satellites have been processed.

The location of each of the GNSS antennas on the three ships is shown in Figure 1. The antennas of R/V Atalante and R/V Meteor are located on the crow's nest, the highest point of the ship, which helps minimizing interference with other scientific

and navigation instruments. The antenna of the R/V Maria S. Merian is located on the higher observation deck below the crow's nest and below the main radar antenna. We will see later that this position has a direct impact on the quality of the measurements carried out with this antenna.





Figure 2 shows the routes followed by the three ships during the campaign. R/V Atalante started its cruise from the Guadeloupe island on 18 January 2020 (day 18) and headed southward until 31 January (day 31) and then back North until reaching Guadeloupe on 21 February (day 52). It mostly operated in the larger ocean area South of Barbados and in the North Brazil Current eddy corridor (the so-called *Boulevard des Tourbillons*). It also conducted operations in the Trade-wind Alley after leaving Guadeloupe on route to the North Brazil Current and prior to returning to Guadeloupe. R/V Maria S. Merian operated between 18 January (day 18) and 19 February (day 50) in the same area as R/V Atalante, South of Barbados and further East, where strong mesoscale ocean eddies are generally present. It also performed measurements in the *Tradewind Alley* while cruising in and out of Bridgetown, Barbados, during the campaign. R/V Meteor operated between 18 January (day 18) and 20 February (day 51) upwind of BCO and of the aircraft operations area. Main cruise track consisting of a 2-day, cross-wind, race-track patterns across the *Tradewind Alley*; its operations were taking place within the *Tradewind Alley* between 12.5 and 14.5N, along the 57.25 W meridian. R/V Maria S. Merian had an encounter with both R/V Atalante and R/V Meteor in the course of the campaign.

The diagram in Figure 3 represents the availability of raw data from each of the three R/Vs. The data acquired by R/V Maria S. Merian have many interruptions (297 over the periods, amounting for about 40 h). These interruptions are mainly due to voluntary cuts to save telecommunication bandwidth for higher priority experiments requiring high data rates. R/V Atalante and R/V Meteor carried out quasi-continuous measurements over the whole period (6 interruptions resulting in a loss of 1 h of data for R/V Atalante, 7 interruptions for a loss of 2 min for data for R/V Meteor).

Table 2 presents some GNSS data quality diagnostics obtained with the translation, editing, and quality check (TEQC) software (Estey and Meertens, 1999). The software operates on the receiver independent exchange (RINEX) observation files. The average number of daily observed GPS satellites was close to the maximum value (currently 31), although the mean value was lower (29) for R/V Maria S. Merian, with a higher standard deviation (4). The parameters $MP1$ and $MP2$ are a measure of the multipath strength embedded in the GNSS phase measurements on frequency carriers L1 and L2. The values remains quite low for R/V Meteor and R/V Atalante which indicates that the antennas on these ships were almost not disturbed by their environment thanks to their location atop the other scientific and navigation antennas (see Figure 1). For R/V Maria S. Merian the multipath values are much higher confirming that the GNSS antenna was disturbed by its surroundings (e.g. reflections on metallic structures, pick up of radar signals). The percentage of daily observations is about 90% of the expected quantity for both R/V Atalante and R/V Meteor. This percentage decreases to 68% for R/V Maria S. Merian, as expected from Figure 3. The last indicator is the ratio of the number of observations per cycle slip (a cycle slip happens when a carrier phase, L1 or L2, is lost). The larger this ratio, the higher the quality of the observations. A typical value of a few hundreds or more is expected. Again, R/V Atalante and R/V Meteor data exhibit a much better quality than R/V Maria S. Merian data.

From these diagnostics, it is to be expected that the quality of GNSS measurements acquired on R/V Atalante and R/V Meteor are adequate to retrieve accurate IWVs. On the other hand, the errors in the position and IWV estimates are expected to be much higher for the R/V Maria S. Merian GNSS data, as confirmed in the following sections.





## 2.2 GNSS data processing

The GNSS observations were initially processed with the GIPSY-OASIS II v6.4 (herefater GIPSY) software in kinematic PPP mode (Zumberge et al., 1997) using standard options similar to the static mode used in Bock et al. (2020a). The software uses the Jet Propulsion Laboratory (JPL) fiducial-free and high rate (30 s) final products 3.0 for satellite orbits and clocks. The data

were analysed in a 30 h window centred on noon of each day from which the 00-24 h parameters were extracted to avoid edge effects. Second order ionosphere correction was used. Phase ambiguities were fixed using the wide lane phase biases computed by JPL as part of their processing of the global GNSS network (Bertiger et al., 2010). The kinematic mode estimates receiver position, clock offsets, ZTDs and horizontal gradients simultaneously for each epoch at a rate of 30 s. No constraint was applied to positions between consecutive epochs.

Tropospheric delays were modelled by time-varying zenith components and horizontal gradients. The zenith components include the Zenith Hydrostatic Delay (ZHD) and the Zenith Wet Delays (ZWD) which represent the contributions of dry air and water molecules, respectively, in the atmospheric column (Bevis et al., 1992). The projection of the zenith delays into the direction of the GNSS satellites is done using the Vienna Mapping Function 1 (VMF1) (Boehm et al., 2006). The projection of the gradient parameters is done using the Bar-Sever et al. (1998) mapping function. ZHD was only corrected a priori while

ZWDs and horizontal gradients were modelled as random walk processes corrections to the a priori values estimated during the data processing with a 30 s time resolution. The random walk process parameters were fixed as in the static mode to 5 mm h$^{-1/2}$ and 0.5 mm h$^{-1/2}$ for ZWDs and gradients, respectively. The a priori values for ZHD and ZWD and the coefficients for the mapping functions were extracted from the Technische Universität Wien (TU-Wien) data base (https://vmf.geo.tuwien.ac.at/). These values are computed from the 6-hourly ECMWF operational analyses by TU-Wien and are distributed on a global

2°×2.5° latitude-longitude grid. In order to take into account the effect of ship along-track displacements on these parameters, a pre-processing was carried out in order to obtain an approximate trajectory. A priori ZHDs and ZWDs were then calculated using a filtered version of this first trajectory every 30 s using a 1-hour median filtering. The mapping functions parameters were calculated only for the average daily positions but were temporally interpolated from the 6-hourly sampling to 30 s.

Two other processing parameters are of importance: the elevation cut-off angle and the observation weighting. In the standard

static processing, we used a 7° cut-off angle and uniform phase observation weighting of 10 mm. The choice of these parameters results from a compromise between including low elevation observations that help decorrelate position and ZTD estimates (this is especially important in kinematic mode where both parameters are estimated at every epoch) and rejecting low elevation observations which are prone to multipath errors. We tested several variants of these parameters and noticed that they had a small but significant impact on the position and ZTD estimates for R/Vs Atalante and Meteor and a very large impact on the

results from R/V Maria S. Merian. The results for the latter were actually very bad as anticipated in the previous Section, and after testing unsuccessfully several other processing options (especially the arc duration for satellite tracking and the ambiguity fixing strategy) we decided to test several other processing software packages.

The SPARK software, available as an online GNSS processing tool of the Canadian Spatial Reference System Precise Point Positioning Service (CSRS-PPP) of Natural Ressources Canada (https://webapp.geod.nrcan.gc.ca/geod/tools-outils/ppp.php,



Banville et al. (2018)) provided the best solution with R/V Maria S. Merian data. The analysis strategy with that software is very similar to that used with GIPSY, namely: kinematic PPP mode with ambiguity resolution, VMF1 modeling for mapping functions, a priori ZHD and ZWD data from TU-Wien, and 30-h processing window. Differences between the softwares concern satellites orbit and clock products, as SPARK uses the International GNSS Service (IGS) final products, and the random walk parameters which are fixed to 3 mm h$^{-1/2}$ and 0.1 mm h$^{-1/2}$ for ZWDs and gradients, respectively. The elevation cut-off

angle in SPARK is fixed to 7.5° and the observation weighting is not specified. However several tests conducted with GIPSY showed that SPARK and GIPSY results for R/Vs Atalante and Meteor agreed best when GIPSY included a $1/\sqrt{sin(elev)}$ weighting. The main disadvantage of the SPARK online service is the impossibility of changing the processing parameters. Nevertheless, the results of SPARK for R/V Maria S. Merian GNSS data remained largely superior to those obtained with GIPSY. It is worth noting that this is the first time in our 15 years of experience in GIPSY that it actually fails to converge

towards an acceptable solution. The problem in the GIPSY processing with the data from R/V Maria S. Merian was identified in the data editing module, which is an upstream processing step, where many observations were deleted because of too many cycle slips. We believe that the main difference is that SPARK uses more modern and efficient data editing and processing algorithms. It is worth noting that the GIPSY software has recently evolved into a new software called GipsyX (Bertiger et al., 2020) which uses more state-of-the-art data editing and processing compared to GIPSY. It is likely that GipsyX can resolve the

problem encountered by GIPSY with the R/V Maria S. Merian data and produce solutions close to those of SPARK. This new software will be tested in the near future.

Regarding the elevation cut-off angle value and observations weighting tests with GIPSY, we noticed that switching from 7.5° (taken identical to SPARK) to 3° changed the mean height estimates for R/Vs Atalante and Meteor by 2.3 mm and 3.6 mm, respectively, and mean ZTDs in a consistent way with a factor of -3.5 approximately. Similarly, the comparison of two

solutions with and without observation weighting (uniform vs. $1/\sqrt{sin(elev)}$) highlighted a difference in the mean height of 5.8 and 15.2 mm for the two R/Vs and consistent differences in mean ZTDs. Such changes are symptomatic of the presence of low-elevation errors due to multipath for instance. The slightly larger variations for R/V Meteor suggest that the data from this R/V are more impacted by multipath errors. The final GIPSY processing options that we retained were thus motivated by the reduction of multipath errors. The cut-off angle was therefore fixed to 7.5° and a down-weighting of low elevation angle

observations was applied. Another advantage of this choice is that the GISPY processing options were consistent with those of SPARK that were used for processing the R/V Maria S. Merian.

## 3 Comparison of processing software results

### 3.1 Formal errors

The first characterization of the processing results was carried out by analyzing the formal errors of the three-dimensional

positions and ZTD estimates. Figure 4 shows the temporal evolution and the histograms of the formal errors for the two processing software packages and the three R/Vs, and Table 3 reports the respective percentile values. Two features stand out from the plots: the shift towards higher values for the SPARK software results and the very large scatter of the R/V Maria S.



Merian results for both software. The shift is mainly linked to the differences in parameterization of the two software (e.g. weighting of measurements, random walks) and input data (e.g. orbit and clocks products). The larger scatter for R/V Maria S.

Merian data is explained by the lower data quality leading to more outliers which are associated with larger formal errors.

The GIPSY results for R/Vs Atalante and Meteor show median formal errors around 25 mm on positions and 1 mm on ZTDs, with 99[th] percentile values around 40 mm for positions and 1.1 to 1.4 mm for ZTDs. The SPARK results are higher by a factor of ∼1.5 for positions and 2 for ZTDs, for both R/Vs. For R/V Maria S. Merian, the median values of formal errors are globally higher compared to the other two R/Vs and the ratio of percentiles between processing software is not constant.

The 99[th] percentile value of position error with SPARK is about 85 cm while it exceeds 100 m with GIPSY which reveals a huge instability in the GIPSY retrievals. Contrary to the position errors, ZTD errors remain small thanks to the constraint in variability imposed by the random walk model, with 99[th] percentile values of 5.6 mm for GIPSY and 16.4 mm for SPARK.

### 3.2 Data screening

The analysis of the distribution of formal errors helped to set the range limits for the post-processing data screening in order to

reject outliers in the ZTD and position estimates (Bock et al., 2020b). Due to the different statistical properties observed in the results discussed above, different thresholds were adopted for the two softwares and the three R/Vs:

- For R/Vs Atalante and Meteor, we observed for both processing a dip in the histogram of the formal errors of positions around 70 mm and 90 mm for GIPSY and SPARK, respectively. Histograms of the formal errors of ZTD did not emphasize any discontinuities. So, for GIPSY estimates, we set the range check upper limit for the formal errors of positions to

70 mm, which led to a rejection of 0.004 % (4 points out of $10^5$) for R/V Atalante and 0.05 % (49 points out of $10^5$) for R/V Meteor. For SPARK estimates, we set the limit for formal errors of positions to 90 mm, which led to no rejection (0 point) for R/V Atalante and a rejection of 0.09 % (85 points) for R/V Meteor. With these range limits, the number of rejections for both software were fairly consistent.

- For R/V Maria S. Merian, the histograms of formal error of positions were more continuous, and only a small dip was

observed around 70 cm for the GIPSY solution. The histograms of formal errors of ZTDs present a dip around 7 mm for GIPSY and 13 mm for SPARK. For GIPSY, upper limits for the formal errors of position and ZTD were therefore set to 70 cm and 7 mm, respectively, which led to a rejection of 6.7 % (5261 points out of $7.510^4$). For SPARK, the limit for formal errors of positions was set 90 cm (i.e. in the same proportion as for the other ships) and the upper limit for the formal errors of ZTD was set to 13 mm, which led to a rejection of 2.82 % (2177 points).

### 3.3 Comparison of position and ZTD estimates

Table 4 gives some statistics on the results from the two software packages after the screening. The average number of satellites used per epoch for R/Vs Atalante and Meteor is nominal (around 10) and consistent between softwares. This is not the case for R/V Maria S. Merian for which this number is much smaller for the GIPSY processing (5.6) although slighter better for SPARK (8.1). As previously mentioned, these numbers suggest that a lot of data were edited by both software, which in the





case of GIPSY become very small and make the solution unstable. Figure 5 compares the height and ZTD estimates for R/V Maria S. Merian from both software from which the instability of the GIPSY solution is obvious.

The other statistics of Table 4 show that both softwares are able to estimate nearly the same number of height and ZTD parameters for R/Vs Atalante and Meteor, although the number of estimates is smaller for SPARK than for GIPSY. The height and ZTD estimates are fairly consistent between software for R/Vs Atalante and Meteor but much larger for R/V Maria S.

Merian despite the outlier screening.

Finally, we decided to keep the GIPSY solutions for R/Vs Atalante and Meteor for the main reason that we have access to more processing output parameters which may be useful for further investigations. For the R/V Maria S. Merian, the SPARK solution was kept because it is obviously of higher quality.

### 3.4   Vertical positioning evaluation

For the assessment of the vertical component of the final estimated positions, we converted the ellipsoidal heights to geoid heights using EGM2008 (Pavlis et al., 2012), which were then compared to sea surface heights derived from operational ocean model products. The mean sea surface height was taken from CNES_CLS2015 model (Pujol et al., 2018), the ocean tides from the FES2014b model (Lyard et al., 2016) and the barometric correction was derived from mean sea level pressure from ERA5 (Hersbach et al., 2020). GNSS height were also corrected from crust deformations using IERS conventional models (Petit and

Luzum, 2010) for solid earth tides and ocean tide loading derived from FES2014b.

Figure 6 shows the differences between the modelled and GNSS-estimated geoid heights for the three R/Vs. Note that since the draught between the antenna reference point and the waterline of the ships is not precisely known (and is subject to variations of several tens of centimetres during the cruise), the differences were corrected for the median antenna heights. As a result, the mean differences for all three ships were insignificant. The GNSS time series were also smoothed using a

5-min median filter in order to reduce fast heave movements of the ships. Standard deviations of the GNSS height errors for the filtered data are on the order of 20 cm for R/V Meteor and R/V Atalante. These errors are consistent with the error budget described by (Bouin et al., 2009) for sea surface height determination by GNSS and the formal errors of the CNES_CLS mean surface that ranges from a few millimeters up to 10 cm in coastal zones over the area. For R/V Maria S. Merian, the standard deviation reaches 40.8 cm (Figure 6). The position estimates for this ship are much noisier, reflecting the poorer quality of the

GNSS data acquired on this vessel as previously mentioned.

Inspection of the time series in Figure 6 shows a period of larger scatter around 20-22 January 2020. We checked that these errors are not related to the JPL and IGS satellite orbit and clock products by performing a kinematic mode processing of GNSS observations for nearby terrestrial stations. The latter did not show this feature. The impact of a higher speed of ships during this period is also not suspected, since the speed values are of the same order throughout the entire campaign. These

variations are therefore more likely related to the sea state during this period.

Finally, small offsets are observed in the height time series of R/V Atalante at the beginning and end of the campaign, as the vessel is docked. These variations are probably due to the lower performance of the mean sea surface model in the coastal waters of Guadeloupe.





## 4  IWV evaluation

### 4.1  GNSS IWV retrieval

As mentioned previously, during the data processing, the ZTD was modelled with 2 components as follows:

$$ZTD = ZHD + ZWD \tag{1}$$

After the processing, we needed to extract the ZWD using a precise estimate of the ZHD. In this work, ZHD was computed from mean sea level pressure extracted from ERA5 reanalysis with a horizontal resolution of 0.25° and temporal sampling of 1 h (Hersbach et al., 2020) using the modified Saastamoinen formula (Saastamoinen, 1972) proposed by Bosser et al. (2007). The ZHD estimates were corrected for the height difference between the GNSS antenna and the mean sea level using the following formula (Steigenberger et al., 2009; Boehm and Schuh, 2013) which was shown to be adequate for small height differences:

$$ZHD(h_{GPS}) = ZHD(h_{ERA5}) - 10^{-6} k_1 \frac{P(h_{ERA5})}{T(h_{ERA5})} \cdot \frac{g_{h_{ERA5}}}{g_{atm}} \cdot (h_{GPS} - h_{ERA5}) \tag{2}$$

where $T(h_{ERA5})$ and $P(h_{ERA5})$ are the mean temperature and pressure from ERA5 between the GNSS antenna and the ERA5 surface, $g(h_{ERA5}) = 9.8062 \, \mathrm{m\,s^{-2}}$; $g_{atm} = 9.7840 \, \mathrm{m\,s^{-2}}$ is the approximated gravity of the center of mass of the atmosphere (Boehm and Schuh, 2013); and $k_1$ is a refractivity constant for dry air.

The final GNSS ZWD estimate was obtained as:

$$ZWD(h_{GPS}) = ZTD - ZHD(h_{GPS}) \tag{3}$$

and the IWV was converted from $ZWD(h_{GPS})$ following:

$$IWV(h_{GPS}) = \kappa(T_m) \times ZWD(h_{GPS}) \tag{4}$$

where $\kappa(T_m)$ is a semi-empirical function of the weighted mean temperature $T_m$ (Bevis et al., 1992):

$$\kappa(T_m) = \frac{10^6}{R_v(k_2' + \frac{k_3}{T_m})} \tag{5}$$

where $R_v = 461.5 \, \mathrm{J\,K^{-1}\,kg^{-1}}$ is the specific gas constant for water vapour, and $k_2'$ and $k_3$ are refractivity constants for the water molecule. In this work we used the refractivity constants updated by Bock et al. (2020b) and the $T_m$ estimates provided by TU-Wien on the same global grid as the a priori ZHD and ZWD products used for GNSS processing. We interpolated the $T_m$ values at each position and time for which the GNSS ZWD estimates were available. The final GNSS IWV estimates were retrieved at a resolution of 30 s for all three R/Vs.

### 4.2  IWV comparisons with ERA5 and MODIS

#### 4.2.1  ERA5 IWV pre-processing

The ERA5 reanalysis IWV data were provided by the Copernicus service (Hersbach et al., 2020) with a horizontal resolution of 0.25° and temporal sampling of 1 h. We first extrapolated the IWV data from the ERA5 model surface to the height of the





GPS antenna by using the empirical formulation first proposed by (O. et al., 2005) and used by (Parracho et al., 2018):

$$IWV_{ERA5}(h_{GPS}) = IWV_{ERA5}(h_{ERA5}) \times \left[1 - 4 \times 10^{-5} \times (h_{GPS} - h_{ERA5})\right] \tag{6}$$

Where $h_{ERA5}$ and $h_{GPS}$ are respectively the geoid heights of the ERA5 grid points and the GNSS antenna; $IWV_{ERA5}(h_{ERA5})$ are the ERA5 IWV values at the grid points; $IWV_{ERA5}(h_{GPS})$ are the extrapolated values at GNSS antenna height.

The final 1-hourly ERA5 IWV values to be compared with GNSS retrievals were computed by bilinearly interpolating the values from the four ERA5 grid points surrounding the GNSS antenna.

### 4.2.2   MODIS IR IWV pre-processing

MODIS IWV retrievals used here are based on clear sky, nighttime and daytime infra-red (IR) products MYD05 and MOD05, collection 6, from the Aqua and Terra satellites, respectively (King et al., 2003). The spatial resolution of MODIS IR products is 5 km. We only used data for which the "Quality_Assurance_Infrared" flag was set to "Useful and Good". For each pass of the Aqua and Terra satellites over the EUREC[4]A domain, the closest pixel representing a valid IWV within a 20 km radius area around each vessel was considered. Furthermore, the time difference between the satellite measurement and the GNSS

measurement was imposed to be less than 15 s.

Note that the MODIS Near-IR IWV product is known to be of higher quality than the IR product but it is only available during daytime, over infra-red reflective surfaces such as clear land, clouds, and oceanic areas in the condition of Sun glinter. The latter condition is very restrictive as only very few observations were found be valid over our study area. For this reason we only used the IR product here.

Several studies have evaluated the MODIS IR IWV product by comparison with ground-based measurements from radiosondes and ground-based water vapor radiometer. They found a RMS difference between 5 and 6 $\mathrm{kg\,m^{-2}}$ in cloud free conditions (Liu et al., 2015; Ferrare et al., 2002). The accuracy of the MODIS IR IWV products is less than expected from the ERA5 reanalysis but it still provides an independent observational source of evaluation for the GNSS retrievals.

### 4.2.3   Comparison results

Figure 7 shows the IWV time series and the differences for the three data sets and the three R/Vs. The agreement between GNSS and ERA5 is fairly good for R/Vs Atalante and Meteor. Both datasets are consistent in depicting the slow temporal variations of IWV during the cruises of both ships. The differences for the shorter time variations rarely exceed $\pm$ 5 $\mathrm{kg\,m^{-2}}$ but in general, the rapid variations are more peaked in the GNSS series. For R/V Maria S. Merian, the agreement is not as good, with IWV differences often exceeding the level of $\pm$ 5 $\mathrm{kg\,m^{-2}}$. Nevertheless, the slow variations in IWV are still

properly retrieved by GNSS in spite of discontinuities in the time series. For the three R/Vs, the agreement with MODIS IR IWV data is globally not as good as with ERA5 IWVs, with much more scattered results, and IWV differences with respect to the GNSS estimates from R/Vs Atalante and Meteor often exceeding the level of $\pm$ 5 $\mathrm{kg\,m^{-2}}$.

Interestingly, the IWV time series for R/V Meteor is more concentrated around 30 $\mathrm{kg\,m^{-2}}$ while the other two R/Vs sampled higher IWV contents during their southward excursions in the Tropics (e.g. up to 50 $\mathrm{kg\,m^{-2}}$ around day 28). Correlated





variations in IWV are also observed by all three ships in different periods as a result of the large-scale atmospheric circulation (e.g. the increasing IWV trend between days 40 and 46).

Statistics of IWV differences between GNSS, ERA5 and MODIS_IR are given in Table 5. The mean difference between GNSS and ERA5 is negative for R/V Atalante (-1.62 $\mathrm{kg\,m^{-2}}$), meaning that GNSS over-estimates IWV compared to ERA5, while the difference with R/V Meteor is positive (+0.65 $\mathrm{kg\,m^{-2}}$). These biases can be seen in the time series of Figure 7 where

the IWV differences for R/V Atalante are mainly below zero while they are more centered on zero for R/V Meteor. The origin of these biases is unclear, and may partly be due to a small differential bias in the ZTD estimates from the two ships (possibly connected with different multipath effects, see the discussion in the previous section). However, it was also observed in past studies that ERA-Interim and ERA5 have a general dry bias in the tropics (Parracho et al., 2018) and in the Caribbeans in particular (Bosser and Bock, 2020). The latter study evidenced that ERA5 had a bias of -1 to -2 $\mathrm{kg\,m^{-2}}$ compared to terrestrial

GNSS stations in the Caribbeans, while this bias was close to zero in the more northern domain of the Caribbeans (close to 18°N). The standard deviations of differences are around 2.3 $\mathrm{kg\,m^{-2}}$ and the correlation coefficients are above 0.89 for both R/V Meteor and above 0.95 for R/V Atalante.

As expected, the agreement with ERA5 is not as good for R/V Maria S. Merian, both in terms of mean (+2.82 $\mathrm{kg\,m^{-2}}$) and standard deviation (> 5 $\mathrm{kg\,m^{-2}}$), with minimum and maximum differences in excess of ±13 $\mathrm{kg\,m^{-2}}$. The correlation

coefficient is also smaller (<0.8) than for the other two ships.

The statistics of comparisons with respect to MODIS_IR are worse than for ERA5, although consistent with the accuracy of this MODIS product over the ocean. GNSS IWV retrievals are dry-biased with respect to MODIS products for both R/V Atalante (-4.98 $\mathrm{kg\,m^{-2}}$) and R/V Meteor (-2.25 $\mathrm{kg\,m^{-2}}$) and standard deviations are in excess of 3.3 $\mathrm{kg\,m^{-2}}$. The correlation coefficient is also smaller for the two ships compared to ERA5, with a value significantly lower for R/V Meteor. Concerning

R/V Maria S. Merian, the statistics of the comparison with MODIS are better than for ERA5, exception made of the correlation coefficient. GNSS IWV retrievals are slightly moist-biased with respect to MODIS products for R/V Maria S. Merian (+0.08 $\mathrm{kg\,m^{-2}}$). Nevertheless, indicators such as the standard deviation, RMS, minimum and maximum differences all suggest that the statistical comparison with MODIS is worse for R/V Maria S. Merian than for the other two ships.

From the statistical analysis between GNSS and ERA5, we may estimate the differential bias between R/V Atalante and

R/V Meteor to ∼2.2 $\mathrm{kg\,m^{-2}}$ on average. The differential bias with respect to ERA5 between R/V Maria S. Merian and R/V Atalante amounts to 4.4 $\mathrm{kg\,m^{-2}}$, and this bias reduces to 2.2 $\mathrm{kg\,m^{-2}}$ between R/V Maria S. Merian and R/V Meteor. The differential biases with respect to MODIS are consistent.

### 4.3 IWV comparisons with ground-based GNSS stations

The IWV estimates from the three ships were also compared to the IWV data retrieved from permanent stations operated

in the Caribbeans and the two GNSS stations (BCON and BCOS) installed at the BCO for EUREC[4]A (Bock et al., 2020a). The ground-based IWV estimates were height-corrected using Eq. 6 and the comparisons were performed when the distance between the ships and the CORS was smaller than 20 km and the height difference smaller than 100 m. Statistics were computed



when at least 20 data points were available. Eight stations were available for comparison with R/V Atalante, two for R/V Maria S. Merian, and two for R/V Meteor.

As an example, Figure 8 compares the IWV data for a short period when the R/V Atalante passed close to the BCO. The upper panel of Figure 8 shows the IWV time series from which it can be seen that the temporal variations between the ship-borne and ground-based GNSS measurements are highly consistent, especially when the ships get closer than 20 km (variations are then well within $\pm 1\ \mathrm{kg\,m^{-2}}$ on middle panel). Comparatively, ERA5 shows large spurious variations, with a marked wet bias between 18:00 and 20:00 UTC (day 47) and a persistent dry bias between 14:00 and 00:00 UTC (day 48).

The statistics of the differences of all IWV retrievals with respect to the BCON station are summarized in Table 6. A difference of -0.02 $\pm$ 0.66 $\mathrm{kg\,m^{-2}}$ is observed between R/V Atalante and BCON when the distance is smaller than 20 km, which increases to 0.81 $\pm$ 1.31 $\mathrm{kg\,m^{-2}}$ when distances up to 100 km are included. ERA5 has a mean bias of -0.81 $\mathrm{kg\,m^{-2}}$ and a standard deviation of difference of 1.79 $\mathrm{kg\,m^{-2}}$. The MODIS comparison is not significant as only three points of comparison are available over this short period. The difference between the two ground-based GNSS stations is -0.10 $\pm$ 0.25 $\mathrm{kg\,m^{-2}}$. This

is an estimate of the internal precision of the technique which is obviously very good.

    The mean and standard deviation of IWV differences from the three ships with all encountered ground-based GNSS stations are plotted as a function of time in Figure 9. The R/V Atalante shows an overall negative bias, meaning that the ship-borne IWV estimates are larger than the ground-based station results, while the R/V Meteor results are of opposite sign. The differential bias between R/V Atalante and R/V Meteor shown in Figure 9 is seen to be $\approx 2$ $\mathrm{kg\,m^{-2}}$ on average. This result is fairly

consistent with the differential bias between the to ships derived from the ERA5 and MODIS comparisons discussed in the previous sub-section, although the time series are largely under-sampled here due to the distance restriction between the ships and ground-based stations. During two periods (day-of-year 19-20 and 49-50), the R/V Atalante crosses a group of four stations in Guadeloupe (marked by dashed-line rectangles in Figure 9). For both periods the bias is more negative compared to BCON and MAGT, located slightly south of the main Guadeloupe island. It can also be noticed that during the second period the

biases are increasingly more negative with time, this may be connected with the ship entering the harbor where more multipath effects are to be expected. For R/V Maria S. Merian the biases are much larger although the standard deviations remain close to those observed for the other comparisons.

### 4.4   IWV comparisons during ships encounters

During the campaign, the R/V Maria S. Merian met with the other two R/Vs several times: three times with R/V Atalante (for a

total duration of about 25 h and 2977 points of comparison) and eight times with the Meteor (for a total duration of about 70 h and 8342 points of comparison). Table 7 shows the mean IWV differences between the ships which can be compared to the differential biases discussed above using ERA5 as a common reference. Here, the bias between R/V Atalante and R/V Maria S. Merian amounts to 5.69 $\mathrm{kg\,m^{-2}}$ and the bias between R/V Meteor and R/V Maria S. Merian reduces to 4.11 $\mathrm{kg\,m^{-2}}$. The agreement with the differential biases found with respect to ERA5 is quite poor (4.4 $\mathrm{kg\,m^{-2}}$ and 2.2 $\mathrm{kg\,m^{-2}}$, for these two

comparisons, respectively). The reason is that the R/V Maria S. Merian has a varying bias component (Figure 7) combined with the fact that these biases are computed over different sample periods. Similarly, the differential bias between R/V Atalante





and R/V Meteor using R/V Maria S. Merian as a common reference amounts to $5.69 - 4.11 = 1.58 \, \mathrm{kg \, m^{-2}}$ which is in poor agreement with the differential bias between these ships estimated previously with respect to ERA5 ($2.2 \, \mathrm{kg \, m^{-2}}$). The standard deviation of the differences are very different ($2.28 \, \mathrm{kg \, m^{-2}}$ between R/V Atalante and R/V Maria S. Merian, and $4.10 \, \mathrm{kg \, m^{-2}}$

between R/V Meteor and R/V Maria S. Merian, which again is explained by the varying bias component of the R/V Maria S. Merian IWV retrievals.

## 5  Conclusions

GPS phase measurements from GNSS instruments on board R/Vs Atalante, Maria S. Merian and Meteor operating during the EUREC[4]A campaign have been processed and the position and IWV estimates have been inter-compared and validated.

A thorough data quality check revealed that the data acquired on R/Vs Atalante and Meteor are of higher quality than the data from R/V Maria S. Merian, likely because of the location of the antenna on the latter R/V making it is more prone to multipath errors and signal pickup from other instruments (e.g. radars). Two software packages were used to process the phase measurements in kinematic PPP mode. The GIPSY software which we use usually to process both static terrestrial and kinematic ship-borne measurements failed at analysing the noisy data from R/V Maria S. Merian. The online SPARK software

performed better, but still the results for the R/V Maria S. Merian were of much lower quality than for the other two ships. Nevertheless, we decided to keep the SPARK results for the R/V Maria S. Merian and the GIPSY results for the other two ships. A post-processing screening procedure was also applied to the ZTD estimates in order to remove outliers, mainly in the R/V Maria S. Merian data.

First the ship-borne GNSS results were assessed by through the comparison of the estimated vertical position components

with sea surface height models. The results were quite conclusive for R/Vs Meteor and Atalante while R/V Maria S. Merian data showed significant inconsistencies. Second, the ship-borne IWV estimates were compared to the ERA5 reanalysis. A small negative bias of $-1.62 \, \mathrm{kg \, m^{-2}}$ was found for R/V Atalante, a small positive bias of $+0.65 \, \mathrm{kg \, m^{-2}}$ for R/V Meteor, and a slightly larger bias of $+2.52 \, \mathrm{kg \, m^{-2}}$ for R/V Maria S. Merian. The RMS differences for the three ships amounted to 2.2, 2.7, and $5.75 \, \mathrm{kg \, m^{-2}}$, respectively. The much larger RMS differences for the R/V Maria S. Merian point to the difficulty to retrieve

accurate IWV estimates from the noisy measurements of the GNSS system on this ship. We also compared the ship-borne IWV retrievals to the MODIS infra-red product. The latter is, however, not very accurate over the oceans and we found RMS differences between 4.8 and $6.0 \, \mathrm{kg \, m^{-2}}$. A differential bias was evidenced between R/Vs Atalante and Meteor of $2.27 \mathrm{kg \, m^{-2}}$. We hypothesize that the ship-borne GNSS measurements from R/Vs Atalante and Meteor may have some small local multipath errors as well. The differential bias between R/V Maria S. Merian and the other two R/Vs was estimated to $4.4 \, \mathrm{kg \, m^{-2}}$ (R/V

Atalante minus R/V Maria S. Merian) and $2.2 \, \mathrm{kg \, m^{-2}}$ (R/V Meteor minus R/V Maria S. Merian). However, we emphasize that the GNSS IWV retrievals from the R/V Maria S. Merian include a strongly varying bias component that makes these results rather inaccurate. Comparison with ground-based GNSS IWV data from stations located along the route of the three R/Vs confirm the previous results.



This study highlights the potential of retrieving IWV estimates from GNSS measurements collected from dual frequency
GNSS or GPS navigation systems on board modern research vessels. The strength of this technique lies in the existence of
a large number of ships (not only research vessels) cruising the oceans that are already equipped with such GNSS systems.
Further investigations are currently made for retrieving IWV estimates in near real-time that would be especially useful for
numerical weather prediction.

*Data availability.* The RINEX files containing the GNSS measurements as well as the screened IWV estimates are available for download
at the AERIS web site:

- GNSS RINEX files:

    - R/V Atalante: https://doi.org/10.25326/74 (Bosser et al., 2020d)

    - R/V Maria S. Merian: https://doi.org/10.25326/75 (Bosser et al., 2020e)

    - R/V Meteor: https://doi.org/10.25326/76 (Bosser et al., 2020f)

- GNSS IWV data:

    - R/V Atalante: https://doi.org/10.25326/71 (Bosser et al., 2020a)

    - R/V Maria S. Merian: https://doi.org/10.25326/72 (Bosser et al., 2020b)

    - R/V Meteor: https://doi.org/10.25326/73 (Bosser et al., 2020c)

*Author contributions.* PB carried out the GNSS data analysis and the comparisons. PB, OB, CF analysed the results and co-wrote the article
with contributions from SB and SP.

*Competing interests.* The authors declare that they have no conflict of interests.

*Acknowledgements.* The authors would like to thank the on-board technical staff of Atalante (Génavir), Maria S. Merian and Meteor (Briese
Research) for their support in the preparation of the on-board GNSS configuration and the acquisition of the GNSS data at sea during the
campaign: Michael Maggiulli, Emmerich Reize, Heinz Voigt-Wentzel, Olaf Willms, Reimar Wolf (Briese Research), Hervé Bisquay and
Thomas Peel (Génavir). Stefan Kinne, Bjoern Bruegmann (Max-Planck-Institut für Meteorologie) and Johannes Karstensen (Geomar) are
also acknowledged for providing contacts and support to the project.

This work was supported by the CNRS program LEFE/INSU through the projects GEMMOC and VEGAN.

MSS_CNES_CLS15 was produced by CLS and distributed by Aviso+, with support from CNES (https://www.aviso.altimetry.fr/)

FES2014 was produced by Noveltis, Legos and CLS and distributed by Aviso+, with support from CNES (https://www.aviso.altimetry.fr/)



440     ERA5 data are provided by Copernicus Climate Change Service (C3S) (2017): Fifth generation of ECMWF atmospheric reanalyses of the global climate . Copernicus Climate Change Service Climate Data Store (CDS), *Accessed 2020-05-01*. https://cds.climate.copernicus.eu/cdsapp#!/home



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

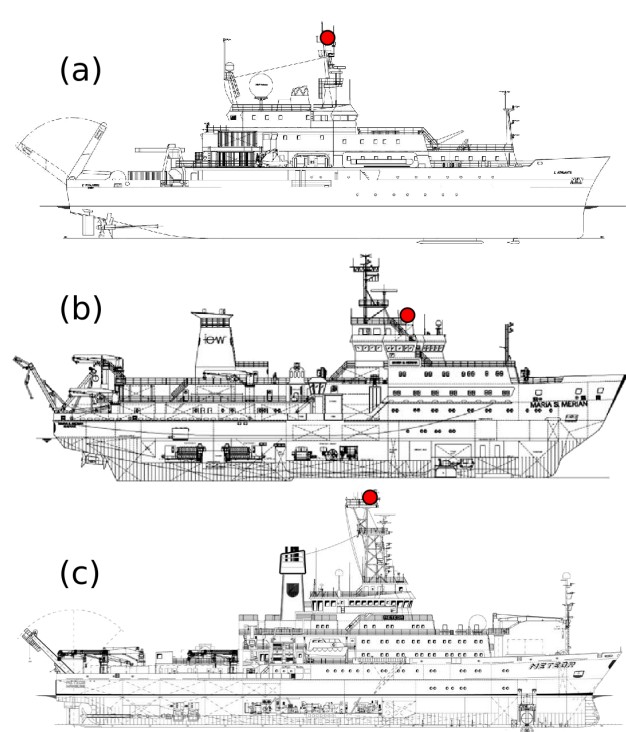

**Figure 1.** Schematic diagram of the location of the used GNSS antennas (red circles) on R/V Atalante (a), R/V Maria S. Merian (b) and R/V Meteor (c) during the EUREC[4]A campaign. Schematic diagrams with courtesy of Genavir and Briese Research.



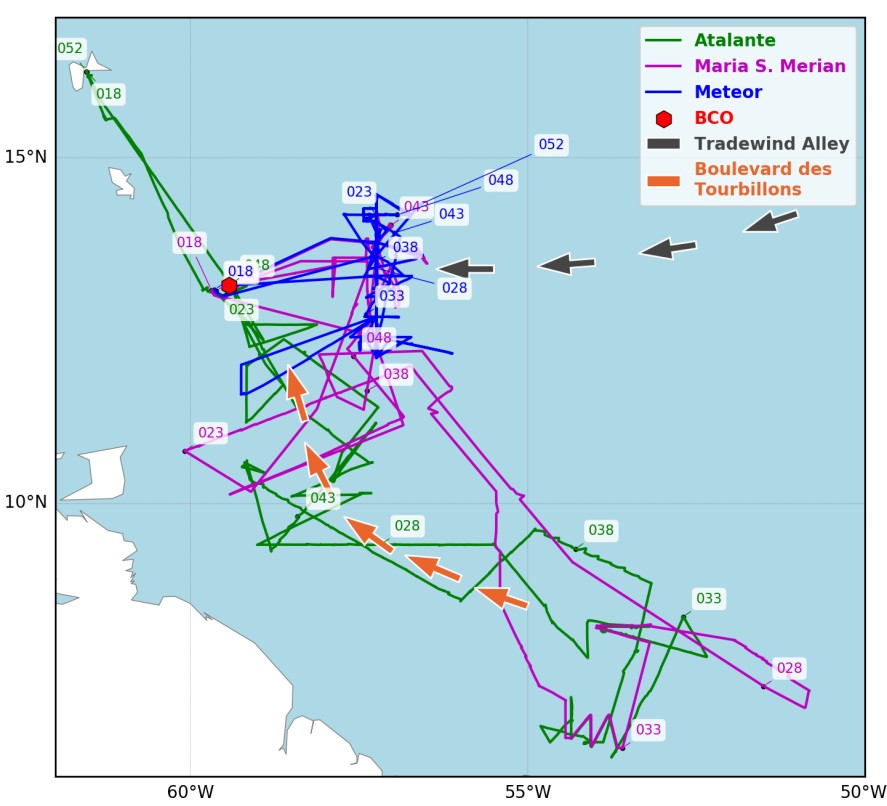

**Figure 2.** Ships trajectory from start to end of the EUREC⁴A campaign. BCO denotes the Barbados Cloud Observatory. Labels indicate vessel passage dates (day-of-year in 2020) and arrows schematic represents the *Tradewind Alley,* (black arrows) and the *Boulevard des Tourbillons* (orange arrows).



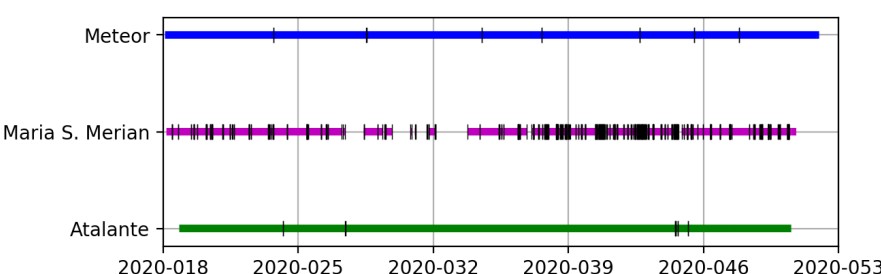

**Figure 3.** Availability of shipborne GNSS measurements (from RINEX files) during the EUREC[4]A campaign. Black vertical ticks denote interruptions in acquisition.



**Figure 4.** Formal errors for position ($\sigma_{POS}$) and ZTD estimates ($\sigma_{ZTD}$) for the three R/Vs, from top to bottom: Atalante, Maria S. Merian, and Meteor. Right: GIPSY processing. Left: SPARK processing. Colored horizontal lines indicate values for 1%, 5%, 10%, 90%, 95% and 99% percentiles.

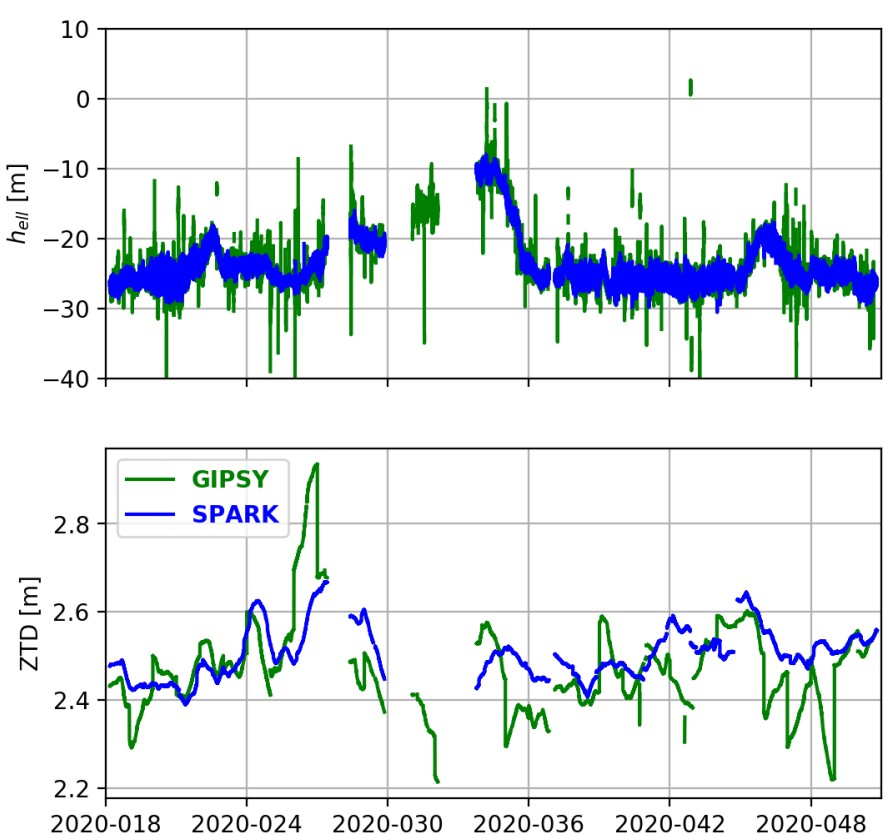

**Figure 5.** Comparisons of ellipsoid height (top) and ZTD (bottom) estimates for R/V Maria S. Merian using GIPSY (green) and SPARK (blue).

**Figure 6.** Geoid height estimates from GNSS and model (top panel) and differences (model - GNSS) (bottom panel) for the three R/Vs: (a) Atalante, (b) Maria S. Merian and (c) Meteor. "Raw" denotes the raw height estimates at a 30 s rate; "Smooth" denotes smoothed height estimates using a 10-min running median. Numerical values indicate mean ± 1 standard deviation.



**Figure 7.** IWV time series from GNSS, ERA5, and MODIS (top panel) and differences with respect to GNSS (ERA5 or MODIS - GNSS) (bottom panel) for the three R/Vs: (a) Atalante, (b) Maria S. Merian and (c) Meteor.
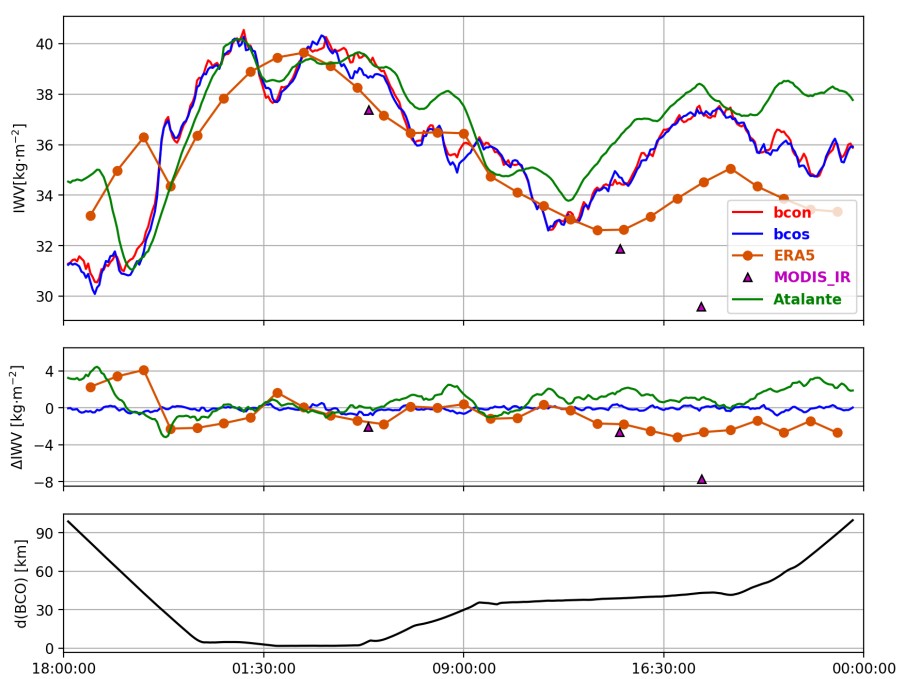

**Figure 8.** IWV time series from R/V Atalante GNSS data, ground-based GNSS data at BCO (station BCON and BCOS), ERA5, and MODIS, as Atalante passes close to the BCO between day-of-year 47 (18:00 UTC) to day-of-year 49 (00:00 UTC). Upper panel represents the IWV time series, the middle panel the differences with respect to BCON and the lower panel the distance between R/V Atalante and BCON station.



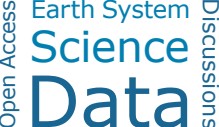

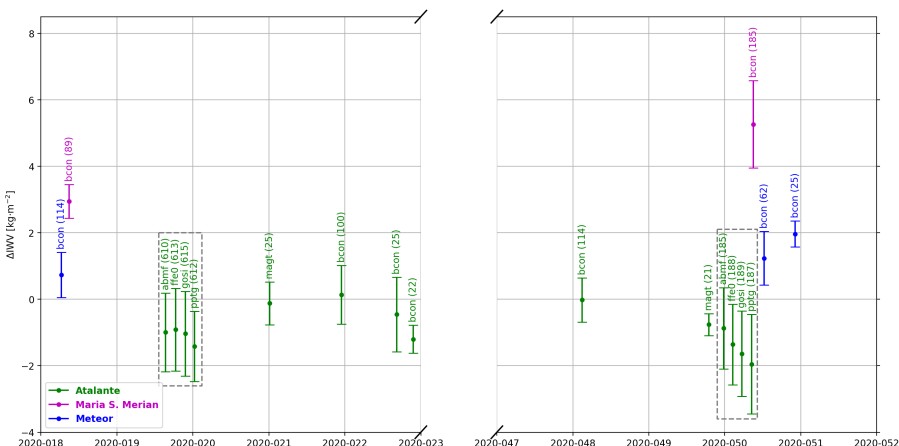

**Figure 9.** Comparisons of GNSS IWV from the three R/Vs to GNSS IWV retrieved from continuously operation reference stations (CORS) when the R/Vs pass close to the stations. The dots represent the mean difference (CORS - R/V) and the error bars the standard deviation of differences. The x-axis represents time. The central period is not displayed because no data could compared as the ship were far offshore. Results enclosed in the two dashed-line rectangles are valid at the same time but are separated in the plot for legibility. The three ships are distinguished by different colors.



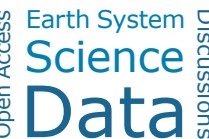

**Table 1.** Summary of shipborne GNSS acquisition systems operated during EUREC[4]A.

| Ship | Receiver | Antenna | Logging |
|------|----------|---------|---------|
| Atalante | Ashtech PROFLEX800 | AeroAntenna Technology AERAT1675_32 | Data server 1h files |
| Maria S. Merian | Trimble SPS855 | Trimble GA530 | Mini PC 1h files |
| Meteor | C-Nav C-Navigator II | Navcom NAVAN2004T | USB Device 1h files |

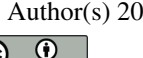



**Table 2.** Quality check diagnostics of GNSS phase observations (GPS measurements only) for the period from day-of-year 18 to 50 in 2020 on board R/Vs Atalante, Maria S. Merian and Meteor. The diagnostics are obtained with TEQC software (Estey and Meertens, 1999): $N_{sat}$ is the average number of satellites per day; $MP1$ and $MP2$ are multipath combinations for L1 and L2 carriers; $\%_{obs}$ is the percentage of complete observations (observed divided by the expected number of observations); $obs/slips$ is the ratio between complete observations and the number of slips. All the numbers are given as daily means $\pm$ 1 standard-deviation.

| Ship | $N_{sat}$ | $MP1$ [m] | $MP2$ [m] | $\%_{obs}$ | $obs/slips$ |
|---|---|---|---|---|---|
| Atalante | 31±2 | 0.32±0.06 | 0.36±0.05 | 90±2 | 671±424 |
| Maria S. Merian | 29±4 | 3.21±0.79 | 3.17±0.83 | 68±5 | 17±4 |
| Meteor | 31±0 | 0.54±0.04 | 0.45±0.02 | 92±1 | 139±22 |

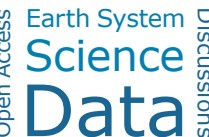

**Table 3.** Percentile values for formal errors of position ($\sigma_{POS}$) and ZTD ($\sigma_{ZTD}$), in mm, for GIPSY and SPARK processing.

| Ship | Software | | $p01$ | $p05$ | $p10$ | $p50$ | $p90$ | $p95$ | $p99$ |
|---|---|---|---|---|---|---|---|---|---|
| Atalante | GIPSY | $\sigma_{POS}$ | 19.8 | 21.2 | 21.9 | 25.5 | 32.0 | 35.1 | 41.7 |
| | | $\sigma_{ZTD}$ | 0.9 | 0.9 | 1.0 | 1.1 | 1.3 | 1.3 | 1.4 |
| | SPARK | $\sigma_{POS}$ | 30.7 | 31.7 | 32.7 | 37.1 | 48.3 | 53.0 | 61.4 |
| | | $\sigma_{ZTD}$ | 1.7 | 1.8 | 1.8 | 2.1 | 2.4 | 2.5 | 2.9 |
| Maria S. Merian | GIPSY | $\sigma_{POS}$ | 27.4 | 30.2 | 32.4 | 46.4 | 273.6 | 1249.0 | $>10^5$ |
| | | $\sigma_{ZTD}$ | 2.0 | 2.1 | 2.2 | 2.7 | 3.9 | 4.4 | 5.6 |
| | SPARK | $\sigma_{POS}$ | 47.7 | 56.6 | 62.1 | 106.9 | 305.7 | 455.2 | 845.1 |
| | | $\sigma_{ZTD}$ | 5.7 | 6.1 | 6.3 | 7.9 | 10.4 | 10.9 | 16.4 |
| Meteor | GIPSY | $\sigma_{POS}$ | 19.9 | 21.1 | 21.8 | 24.6 | 30.9 | 32.8 | 39.2 |
| | | $\sigma_{ZTD}$ | 0.8 | 0.8 | 0.9 | 1.0 | 1.1 | 1.1 | 1.1 |
| | SPARK | $\sigma_{POS}$ | 31.1 | 31.8 | 32.3 | 36.5 | 47.7 | 51.8 | 59.7 |
| | | $\sigma_{ZTD}$ | 1.6 | 1.6 | 1.7 | 1.9 | 2.1 | 2.2 | 2.5 |



**Table 4.** Comparisons of GIPSY and SPARK processing results: $N_{sat}$ average number of satellites per epoch; $N_{ZTD}$ the total number of ZTD estimates available after the post-processing data screening, $\Delta H$ and $\Delta ZTD$ the differences of height and ZTD estimates (SPARK minus GIPSY). Average number of satellites and differences are given as means $\pm$ 1 standard-deviation.

| Ship | $N_{sat}$ (GIPSY) | $N_{sat}$ (SPARK) | $N_{ZTD}$ (GIPSY) | $N_{ZTD}$ (SPARK) | $\Delta H$ [mm] | $\Delta ZTD$[mm] |
|---|---|---|---|---|---|---|
| Atalante | 9.5±1.4 | 10.1±1.4 | 91,235 | 91,173 | -3.4±26.4 | 0.7±4.3 |
| Maria S. Merian | 5.6±1.2 | 8.1±1.3 | 73,256 | 74,925 | -62.2±1624.1 | 29.5±91.2 |
| Meteor | 10.0±1.0 | 10.2±1.3 | 94,690 | 94,649 | 1.6±34.2 | -0.2±5.7 |

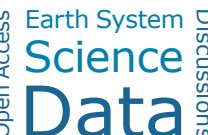

**Table 5.** Comparisons of ERA5 and MODIS IWV with respect to shipborne GNSS estimates: $N_{IWV}$ number of IWV comparisons, mean, standard deviation, RMS, minimum and maximum of IWV differences (ERA5 or MODIS minus GNSS), and correlation coefficient.

| | $N_{IWV}$ | mean diff. [$\mathrm{kg\,m^{-2}}$] | std.dev. diff. [$\mathrm{kg\,m^{-2}}$] | RMS diff. [$\mathrm{kg\,m^{-2}}$] | min / max [$\mathrm{kg\,m^{-2}}$] | corr. coef. |
|---|---|---|---|---|---|---|
| *ERA5 − GNSS* | | | | | | |
| Atalante | 762 | -1.62 | 2.22 | 2.75 | -8.46 / +4.79 | +0.954 |
| Maria S. Merian | 646 | +2.82 | 5.74 | 6.39 | -13.45 / +17.43 | +0.787 |
| Meteor | 789 | +0.65 | 2.35 | 2.44 | -10.73 / +9.60 | +0.891 |
| *MODIS_IR − GNSS* | | | | | | |
| Atalante | 56 | -4.98 | 3.31 | 5.97 | -10.50 / +3.62 | +0.896 |
| Maria S. Merian | 47 | +0.08 | 5.19 | 5.19 | -9.51 / +13.49 | +0.704 |
| Meteor | 64 | -2.25 | 4.25 | 4.81 | -10.98 / +9.60 | +0.590 |



**Table 6.** Comparisons of IWV data from GNSS on R/V Atalante, ERA5, MODIS, and GNSS station BCOS to IWV data from GNSS station BCON from day-of-year 47 (18:00) to day-of-year 49 (00:00): $N_{IWV}$ number of comparisons, mean, standard deviation, and RMS of differences.

|  | $N_{IWV}$ | mean diff. [$\mathrm{kg\,m^{-2}}$] | std.dev. diff. [$\mathrm{kg\,m^{-2}}$] | RMS diff. [$\mathrm{kg\,m^{-2}}$] |
|---|---|---|---|---|
| Atalante ($d < 20$km) | 114 | 0.20 | 0.66 | 0.66 |
| Atalante ($d < 100$km) | 354 | 0.81 | 1.31 | 1.54 |
| ERA5 | 29 | -0.81 | 1.79 | 1.96 |
| MODIS_IR | 3 | -4.11 | 2.54 | 4.83 |
| BCOS | 354 | -0.10 | 0.25 | 0.27 |



**Table 7.** Comparisons of IWV from R/Vs Atalante and Meteor to IWV from Maria S. Merian: $N_{IWV}$ number of comparisons, mean, standard deviation, and RMS of differences.

|  | $N_{IWV}$ | mean diff. [kg m$^{-2}$] | std.dev. diff. [kg m$^{-2}$] | RMS diff. [kg m$^{-2}$] |
|---|---|---|---|---|
| Atalante | 2977 | 5.69 | 2.28 | 6.13 |
| Meteor | 8342 | 4.11 | 4.10 | 5.80 |