# Peer review of "Integrated water vapour content retrievals from ship-borne GNSS receivers during EUREC4A"

_Earth System Science Data, 2020_

## Referee Comment (RC1) · 24 Nov 2020

General comments: This paper describes how to use the GNSS signal obtained on board a going ship to retrieve atmospheric water vapor content, and compares the GNSS IWV retrievals with IWV estimates from the European Center for Medium-range Weather Forecast (ECMWF) fifth ReAnalysis (ERA5), from the Moderate-Resolution Imaging Spectroradiometer (MODIS) infra-red products, and from terrestrial GNSS stations located along the tracks of the ships. The structure of the paper is reasonable, and the description of data processing is very detailed. I think the manuscript is good, and the only flaw is that the paper needs to add some mathematical formulas, which are how to calculate the atmospheric water vapor content.

[Figure]

Specific remarks:

1. Lines 122-124: "The GNSS observations were initially processed with the GIPSY-OASIS II v6.4 (herefater GIPSY) software in kinematic PPP mode (Zumberge et al., 1997) using standard options similar to the static mode used in Bock et al. (2020a). The software uses the Jet Propulsion Laboratory (JPL) fiducial-free and high rate (30 s) final products 3.0 for satellite orbits and clocks." It is best if you can list the key mathematical formulas used in the software below. This way we can more clearly understand the method.

2. Line 249-250: "These variations are therefore more likely related to the sea state during this period." Please explain how the sea surface will affect it. For example, the receiver also receives the reflected signal from the sea surface?

3. The comparison between different water vapor products, the error caused by the scale effect needs to be more explained or discussion.

In general, this manuscript gives a very detailed description of data testing and processing, and did a lot of comparative verification analysis. I suggest that it be published after minor revisions.
* * *

---

## Referee Comment (RC2) · Anonymous Referee #2 · 21 Jan 2021

The manuscript by Bosser et al. describes shipborne GNSS based measurements of integrated water vapor (IWV) during EUREC4A field experiment. The data acquired on three vessels is described in good detail and the data quality evaluated thoroughly and compared to other source of IWV like ERA5 analyses or GNSS station on land. The exposition is easy to follow and covers the right amount of detail. The presented dataset is of great interest for the interpretation of other data sets acquired during EUREC4A and surely deserves publication. The thorough validation ensures that the limitations of some of the data can be taken into account in an adequate manor. I have no major comments, but only minor suggestions for clarification and enhanced readability.

Minor comments:

L103: Please add short explanation what MP1 and MP2 physically mean?

L250: I guess that observational data on the sea state are available for the different R/Vs to support this conjecture. You could be more explicit on this.

L266: The gravity value g_atm depends on position. In simpler models at least on the latitude. The value given seems to be correct for about 15 °N. So the given value is adequate for the measurements described here, but not in general. The Authors should make this clear.

Section 4.1: For completeness and reference the authors should give the values of the three refractivity constants k1, k2, k3 used in the described retrieval.

Eq. 6: There seems to be a unit missing at the scaling constant to make the product with the height difference dimensionless.

L 287: As written it seems that no temporal interpolation between the two nearest ERA5 times has been performed. Is there a reason for omitting this relatively easy step?

L 322ff: The bias of the ERA5 re-analysis during the EUREC4A period is hard to compare with normal times, as a lot of additional WV data (especially the dropsondes from the research airplanes) had been feed into the ECMWF operational system. So one should be very carful in drawing such conclusions. And for the R/V Atalante the bias does not seem to be latitude dependent as it should be according to the statement about the regional dependence of ERA5 in this paragraph.

L347: CORS is not defined here, but only in a figure caption…

L399: by or through, one has to go…

Fig. 3 is mostly of illustrative nature, but nothing quantitative can be concluded. May be left out.

Fig. 4: Annotations and scale number are too small and very hard to read. A clearer annotation of R/V and software would be good. Are the software attributions swapped? In the text it is stated that SPARK has a better filtering algorithm and in the graphs it appears that the data for Maria S. Merian is better filtered by GIPSY

Fig. 6, Fig. 7, Fig. 9 Annotation and scale fonts are far too small!!!

---

## Author Comment (AC1) · 15 Feb 2021

We would like to thank to the reviewers for their constructive comments. We have revised the manuscript accordingly.

We have also corrected some typos and improved wording.

Reviewer 1

1. Lines 122-124: "The GNSS observations were initially processed with the GIPSY-OASIS II v6.4 (hereafter GIPSY) software in kinematic PPP mode (Zumberge et al.,1997) using standard options similar to the static mode used in Bock et al. (2020a).

[Figure]

The software uses the Jet Propulsion Laboratory (JPL) fiducial-free and high rate (30 s) final products 3.0 for satellite orbits and clocks." It is best if you can list the key mathematical formulas used in the software below. This way we can more clearly understand the method.

The processing of GNSS observations involves many mathematical and physical modelling aspects, that cannot be summarized in a few equations. For the basic understanding of GNSS data modelling and analysis we refer the referee to Hofmann-Wellenhof et al. (2008). It is customary in the GNSS application field to indicate the used data processing software and to specify the processing details as we did in Section 2.2.
Hofmann-Wellenhof et al., GNSS - Global Navigation Satellite Systems. Springer, Vienna. https://doi.org/10.1007/978-3-211-73017-1_7, 2008.

2. Line 249-250: "These variations are therefore more likely related to the sea state during this period." Please explain how the sea surface will affect it. For example, the receiver also receives the reflected signal from the sea surface?

The idea is that the sea state can affect the geoid height since waves and swell will induce vertical displacements of the R/Vs (due to roll, pitch, and heave) and thus of the GNSS antennas, even if the speed of the ships remains low. We used the ERA5 product "significant height of combined wind waves and swell" as a proxy of the sea state and found a high degree of correlation between periods of increased scatter of the GNSS vertical component and periods of rough sea state given by the proxy. High waves and swell can therefore be pointed as the origin of large scatter. Figure 6 and its description in text have been updated consequently.
The referee's suggestion of the impact from reflected signals is another possible reason for a scatter in the GNSS height estimates. However this effect is not suspected to be emphasized in case of rough sea conditions (we found no mention in the literature of such an effect).

3. The comparison between different water vapor products, the error caused by the scale effect needs to be more explained or discussion.

We added a special note on spatial and temporal representativeness in Section 4 as follows:
"The GNSS, ERA5, and MOIDS data are time-matched for the comparisons, i.e. the closest GNSS estimate is used for each of the ERA5 or MODIS estimates. The time difference is thus implicitly limited to $\pm$ 15 s. Since, all three data sets provide more or less instantaneous IWV estimates, there is no significant difference in the time scales."
Regarding the spatial scale, GNSS measurements are analysed within a cutoff angle of 7.5 deg above the horizon. Assuming a representative height for water vapor in the atmosphere of about 5 km yields a IWV footprint with a radius of about 35 km around the GNSS antenna. ERA5 has a horizontal resolution of 0.25 deg (25-30 km) which is fairly consistent with GNSS retrievals. On the other hand, MODIS has a horizontal resolution of 5 km. However, for the comparison with GNSS and ERA5, we have considered MODIS pixels within a 20 km radius around GNSS antenna to compute the MODIS-related IWV. Therefore the horizontal scales of the retrievals associated with the different techniques are fairly consistent and no significant spatial representativeness errors are expected.

Reviewer 2

L103: Please add short explanation what MP1 and MP2 physically mean?

A short explanation has been added as suggested : "(interference in the code and phase measurements induced by reflections or scattering by surfaces close to the GNSS antenna)". The computational details are given in the reference quoted in the paper (Estey and Mertens, 1999).
The basic concept is also explained here: https://gssc.esa.int/navipedia/index.php/Multipath

[Figure]

L250: I guess that observational data on the sea state are available for the different R/Vs to support this conjecture. You could be more explicit on this.

As included in the reply to reviewer 1, in order to confirm the impact of sea state on GNSS height time series, we compared the difference of geoid height from GNSS and model with the "significant height of combined wind waves and swell" product from ERA5; we observe a good agreement between the period of large scatter and the period of high waves and swell. High waves and swell can therefore be pointed as the origin of large scatter. Figure 6 and its description in text have been updated consequently.

L266: The gravity value g_atm depends on position. In simpler models at least on the latitude. The value given seems to be correct for about 15degN. So the given value is adequate for the measurements described here, but not in general. The Authors should make this clear.

As both $g_{atm}$ and $g_{ERA5}$ are latitude dependent and vary in very consistent way (Nilson et al., 2013), it is a good approximation to assume that the ratio of these two terms is nearly constant.

Section 4.1: For completeness and reference the authors should give the values of the three refractivity constants k1, k2, k3 used in the described retrieval.

Values have been added as suggested : $k_1 = 0.77643$ K Pa$^{-1}$, $k_2' = 0.22958$ K Pa$^{-1}$, $k_3 = 3752.01$ K$^2$ Pa$^{-1}$.

Eq. 6: There seems to be a unit missing at the scaling constant to make the product with the height difference dimensionless.

Correct. The scaling constant has unit (kg m$^{-3}$). It has been clarified in the text.

L 287: As written it seems that no temporal interpolation between the two nearest

ERA5 times has been performed. Is there a reason for omitting this relatively easy step?

The interpolation of one hourly IWV data to 30 s would introduce correlated errors and bias the comparisons, so we decided to compare ERA5 and GNSS at 1-hourly resolution by matching the times.

L 322ff: The bias of the ERA5 re-analysis during the EUREC$^4$A period is hard to compare with normal times, as a lot of additional WV data (especially the dropsondes from the research airplanes) had been feed into the ECMWF operational system. So one should be very careful in drawing such conclusions. And for the R/V Atalante the bias does not seem to be latitude dependent as it should be according to the statement about the regional dependence of ERA5 in this paragraph.

We agree with the referee and rephrased the sentence:
"However, small dry biases were also reported in ERA5 over the Caribbean Arc compared to ground-based GNSS IWV data during the NAWDEX campaign (Bosser and Bock, 2021) and during EUREC$^4$A (Bock et al., 2021), although they were smaller during the latter, possibly thanks to the additional observations assimilated in the re-analysis during the EUREC$^4$ campaign."

L347: CORS is not defined here, but only in a figure caption...

The CORS acronym has been removed from the manuscript and we refer to "ground-based GNSS stations" instead.

L399: by or through, one has to go...

Corrected.

Fig. 3 is mostly of illustrative nature, but nothing quantitative can be concluded. May be left out.

We believe that this figure is instructive since it clearly illustrates the frequent measurement interruptions with the GNSS antenna of the R/V Maria S. Merian. We therefore like to keep it in.

Fig. 4: Annotations and scale number are too small and very hard to read. A clearer annotation of R/V and software would be good. Are the software attributions swapped? In the text it is stated that SPARK has a better filtering algorithm and in the graphs it appears that the data for Maria S. Merian is better filtered by GIPSY

Figure has been improved.
Indeed the software attributions was swapped. It's fixed now.

Fig. 6, Fig. 7, Fig. 9 Annotation and scale fonts are far too small!!!

Figures have been improved.